# Establishment and application of a dual real-time PCR assay for differential detection of PiAdV-1 and PiAdV-2 among pigeons in China in 2022–2023

Lu Chen[1⊙], Jingjing Wang[1⊙], Yufeng Liu[1,2], Yuteng Chen[1,3], Haiming Wang[1,4], Jinping Li[1], Wenming Jiang[1], Xiaohui Yu[1*], Hualei Liu[1*]

1 China Animal Health and Epidemiology Center, Qingdao, Shandong, China, 2 Qingdao Agricultural University, Qingdao, Shandong, China, 3 Shandong Agricultural University, Taian, Shandong, China, 4 Ningxia University, Yinchuan, Ningxia, China

⊙ These authors contributed equally to this paper.
* yxhui1030@126.com (XY); hualeiliuwy@163.com (HL)

## Abstract

Pigeon virus diarrhea is very common in the clinic, which has caused great damage to the pigeon industry. Pigeon adenovirus constitutes a significant pathogen responsible for pigeon viral diarrhea. Among them, Pigeon adenovirus 1 (PiAdV-1) and Pigeon adenovirus 2 (PiAdV-2) are the most common adenoviruses discovered in pigeon diseases in recent years. Infection with these viruses is common in clinical settings, which makes differential diagnosis more difficult. At present, real-time PCR is one of the most widely used techniques for the identification and detection of pathogens. This study presents the development of a dual real-time PCR assay designed specifically for the detection of PiAdV-1 and PiAdV-2. The dual real-time PCR assay developed in this study demonstrated satisfactory specificity, sensitivity, repeatability, and reproducibility, while showing no cross-reactivity with unrelated pathogens. The limit of detection (LOD) for PiAdV-1 is 94.8 copies/µL, and the LOD for PiAdV-2 is 88.4 copies/µL. This method was then used to analyze 20 liver tissue samples collected from dead pigeons, and the results were verified by genome sequencing. To understand the current epidemic status of pigeon adenovirus in China, the dual real-time PCR method established in this study was used to detect 500 pigeon swab samples collected from 10 provinces in China from 2022 to 2023. The results showed that the positive rates of PiAdV detected by the real-time PCR was 27.00% (135/500). PiAdV-1 single infection was detected in 8 cases (1.6%), PiAdV-2 single infection in 123 cases (24.6%), and PiAdV-1 and PiAdV-2 co-infection in 4 cases (0.8%). This study established a valuable tool for differentiating between PiAdV-1 and PiAdV-2 in practical applications and provided significant insights into the prevalence of these viruses in China.

**Data availability statement:** All relevant data are within the paper.

**Funding:** This study was funded by the National Key Research & Development Program (2022YFD1800600).The funders had no role in study design, data collection and analysis, decision to publish, or preparation of the manuscript.

**Competing interests:** The authors have declared that no competing interests exist.

## Introduction

Adenovirus infection is a common infectious disease in poultry and other farm animals. Adenoviridae viruses are non-enveloped, and the diameter of their icosahedral capsid is approximately 90 nanometers. Their genomes are composed of linear double-stranded DNA and are approximately 25–48 kb in size [1]. The capsid proteins mainly consist of Hexon protein, Penton protein and Fiber protein [2,3], which contains specific antigenic determinants of type, group,and subgroup. According to the latest report of the International Committee on Taxonomy of Viruses (ICTV), the Aviadenovirus genus can be divided into 16 species, including Aviadenovirus leucophthalmi, Fowl adenovirus A-E, Duck adenovirus B, Pigeon adenovirus 1, Pigeon adenovirus 2 [4], Goose adenovirus A, Turkey adenovirus B-D, Psittacine aviadenovirus B-C, and Falcon adenovirus A [5]. Pigeon Adenovirus is a dual-stranded DNA virus lacking a viral envelope [6].

In pigeons, adenoviruses are known to cause two primary disease syndromes. The first is a severe form marked by lethargy and sudden death, with postmortem examination revealing hepatic necrosis. The second presents as a milder condition, commonly associated with young pigeon diseasey, characterized by symptoms such as anorexia, weight loss, diarrhea, and regurgitation [1]. In 1945, a classical PiAdV-1 was first detected in pigeons in Belgium and since then it has subsequently been identified in many parts of the world. Although pigeons of any age can be infected, young pigeons under one year old are more severely affected by PiAdV-1, presenting with vomiting, anorexia and acute watery diarrhoea. The feed intake of sick pigeons decreased significantly, which seriously affected the performance of racing pigeons [7–9]. Between 1990 and 1996, the positive rate for PiAdV-1 was 2.3% in pigeons submitted for etiological testing at Gent University [8]. The PiAdV-2 was detected until 1992 and the first case also occurred in Belgium [10]. PiAdV-2 affects pigeons of all ages and is characterized by sudden death and severe liver necrosis [7,11]. The distribution of the PiAdV-2 was investigated in fecal samples of healthy or young pigeon disease syndrome (YPDS) affected pigeons collected from different lofts between 2008 and 2015. Independent of health status, approximately 20% of young birds and 13% of adult pigeons contained the PiAdV-2 in their feces [5].

The diagnosis of pigeon adenovirus infection mainly relies on laboratory testing. Virus isolation and identification are the most definitive methods for confirming viral pathogenicity, while virus isolation for pigeon adenovirus is very difficult. Conventional PCR detection technology are common detection methods in laboratory diagnosis, while this method is more time-consuming and lower sensitivity compared with real-time PCR assays. The molecular detection methods such as conventional PCR assays and real-time PCR assays are currently the major methods used for the detection of pigeon adenovirus infection. PiAdV-1 and PiAdV-2 are two different species of viruses, with only 54.9% nucleotide sequence homology [5], and the clinical symptoms caused by PiAdV-1 and PiAdV-2 are also different. In order to accurately determine the pathogen, it is necessary to establish a high-throughput detection method that can distinguish the detection of PiAdV-1 and PiAdV-2, so as to facilitate

rapid clinical diagnosis, achieve early detection, early isolation and early treatment of pigeon flocks, thereby reducing the losses of pigeon flocks, and on this basis, grasp the epidemic situation of PiAdVs in pigeon flocks in China.

In this study, a dual real-time fluorescent PCR assay for PiAdV-1 and PiAdV-2 with high specificity and sensitivity was established to provide technical support for the rapid clinical diagnosis of PiAdV-1 and PiAdV-2. Using this method, we conducted an epidemiological survey of two species of pigeon adenovirus on pigeon samples from a total of 10 provinces in China in 2022−2023 to understand the prevalence and distribution of PiAdV-1 and PiAdV-2 in China. At the same time, the representative positive samples were sequenced and the genetic evolutionary relationship was analyzed in order to provide reference for the comprehensive prevention and control of pigeon adenovirus.

## Materials and methods

### Ethics statement

This study was conducted according to the animal welfare guidelines of the World Organization for Animal Health and approved by the Animal Welfare Committee of the China Animal Health and Epidemiology Center. The approval number was DWFL-2023-07.

### Primers and probes

For PiAdV-1 and PiAdV-2, we downloaded at least 19 genome sequences from NCBI for comparative analysis. Primers and probes were designed using Primer Premier 5 software, targeting the most conserved regions of these viruses. The PiAdV-1 probe is labeled with FAM at the 5' end, and BHQ-1 is used as the 3' terminal quencher. The PiAdV-2 probe is labeled with CY5 at the 5' end and the 3' quencher of BHQ-2 (Table 1). All of the primers and probes were synthesized by Sangon Biotech Corporation (Shanghai, China).

### Virus and samples collection

DNA samples of PiAdV-1 and PiAdV-2, Pigeon circovirus (PiCV), Pigeon herpesvirus (PiHV), along with RNA samples of Avian influenza virus (AIV), Newcastle disease virus (NDV), and Pigeon rotavirus A (RVA), were stored at −20°C in the Avian Disease Surveillance Laboratory at the China Animal Health and Epidemiology Center.

Liver tissue was dissected from 20 pigeons that showed clinical signs such as vomiting and diarrhea from ten pigeon farms in Shandong Province. All samples were homogenized in 1.5 mL of phosphate-buffered saline (PBS) supplemented with antibiotics, using a tissue-to-PBS ratio of 1:3. The homogenized samples were then centrifuged at 4°C and 12,000 rpm for 5 minutes to obtain the supernatant.

500 swabs (oropharyngeal swab and cloacal swab of one pigeon mixed in one tube) were randomly collected from apparently healthy pigeons at live bird markets (LBMs) across 10 provinces in China during an active national surveillance program conducted from 2022 to 2023 (Table 5). All samples were homogenized in 1.5 mL of PBS supplemented with antibiotics, followed by centrifugation at 4°C and 12,000 rpm for 5 minutes. The supernatants from these samples were

**Table 1. Primers and probes used in this study.**

| Primer/probes | Sequences (5′—3′) | Genes | Amplicons |
|---|---|---|---|
| PiAdV-1-F1 | GAGGACCTCCAGCAGTTCATC | Hexon | |
| PiAdV-1-R1 | TACGCGTTGGTGGTGTCATC | | 133 bp |
| PiAdV-1-P1 | CAGCTACTTCGAGTTGCGCAACAAG | | |
| PiAdV-2-F1 | TGGGHGAGCTSACYGATCTG | Hexon | |
| PiAdV-2-R1 | AAGCCATGGACAACACRTTC | | 174 bp |
| PiAdV-2-P1 | CATGTATACGAACAACTCGCACAGCAT | | |

then used for RNA/DNA extraction using the Fine Pure Virus DNA/RNA Column Extraction Kit (centrifugal column type) provided by JIFAN BIOTECH (Beijing, China).

### Condition optimization

The concentrations of the primers and probes were optimized. Following this optimization, the real-time PCR reaction was conducted in a total volume of 25 μL, comprising: 12.5 μL of 2 × Probe qPCR Mix MultiPlus, 2 μL of DNA templates, 0.7 μL of each primer (PiAdV-1-F1/R1) at a concentration of 10 μM, 0.9 μL of each primer (PiAdV-2-F1/R1) also at 10 μM, 0.8 μL of each probe (PiAdV-1-P1 and PiAdV-2-P1) at 10 μM, with the remaining volume made up by ddH$_2$O to reach a total of 5.7 μL. Amplification was carried out on the Applied Biosystems QuantStudio 5 platform (Thermo Fisher Scientific, USA), under the following conditions: an initial denaturation at 95°C for 20 seconds, followed by 45 cycles of 95°C for 1 second and 60°C for 20 seconds. The fluorescent signal was measured at the end of the extension phase in each cycle.

### Standard curve generation

Amplified fragments generated with the primer pairs PiAdV-1 F1/R1 and PiAdV-2 F1/R1 were synthesized and subsequently cloned into the pEASY-T5 vector. Using the NanoDrop 2000 spectrophotometer, the concentration of the recombinant plasmid was quantified, and the copy number was calculated as follows: $y(copies/μL) = (6.02 × 10^{23}) × (ng/μL × 10^{-9})/(DNAlength × 660)$. Serial dilutions of the standard plasmid, in 10-fold increments, were then prepared and used as templates to construct the standard curve for the real-time PCR assay.

### Specificity, sensitivity and repeatability

To assess the specificity of the established dual real-time PCR assay, RNA samples from AIV, NDV, and RVA, as well as DNA samples from PiCV and PiHV were utilized. The sensitivity and LOD of the dual real time PCR were determined using 10-fold serial dilutions of standard plasmids.

The repeatability experiment was conducted in triplicate under optimized reaction conditions. It involved three serial 10-fold dilutions for PiAdV-1, ranging from $1.0 × 10^9$ to $1.0 × 10^7$ copies/μL, and for PiAdV-2, from $1.0 × 10^6$ to $1.0 × 10^4$ copies/μL. For each assay, the intra- and inter-group coefficients of variation (CV) were calculated.

### Clinical sample detection

The PiAdV-1 and PiAdV-2 dual real-time PCR assay established in this study was used to detect 20 pigeon clinical samples. The results were compared and analyzed with those of the conventional PCR [5,12] (Table 2) methods and sequencing results commonly used in the clinic.

### Epidemiological surveillance of PiAdV-1 and PiAdV-2 in China from 2022 to 2023

The dual fluorescence quantitative PCR method established in this study was used to detect 500 swab samples collected randomly at LBMs in 10 provinces of China from 2022 to 2023 with Probe qPCR Mix MultiPlus kit. The distribution and positive rate of PiAdV-1 and PiAdV-2 were analyzed.

**Table 2. Primers used in this study.**

| Primes | Sequence (5'-3') | Genes | Amplicons |
|---|---|---|---|
| PiAdV-1-F2 | ATCAACTACGACAACGAAGGC | Fiber-2 | 967 bp |
| PiAdV-1-R2 | CGGTAGAGTTACGGGGAAATT | | |
| PiAdV-2-F2 | GTAACATGAGCGTGCTGTTTG | Hexon | 643 bp |
| PiAdV-2-R2 | CTGAGAAACGAAACCCGAATTG | | |

## Sequencing and phylogenetic analysis

Fifteen pigeon adenovirus strains isolated from different provinces were selected for sequencing and phylogenetic analysis. The Hexon gene of pigeon adenovirus was amplified by conventional PCR [5], and the PCR products were analyzed by 1% agarose gel electrophoresis. The amplified products were then sequenced by Qingdao Ruibo Biotechnology Co., Ltd. Reference strains with different host origins and genotypes were downloaded from the GenBank database (Table 3). Phylogenetic analysis was performed using MEGA 6 software, and the neighbor-joining method was employed to construct the genomic evolutionary tree with 1,000 bootstrap replicates for validation. Additionally, MegAlign software was used for genomic homology analysis.

## Results

### Standard curve of the dual real-time PCR

The dual real-time PCR assay were generated using the concentrations ranged from $9.48 \times 10^9$ to $9.48 \times 10^5$ copies/μL for PiAdV-1 and from $8.84 \times 10^7$ to $8.84 \times 10^3$ copies/μL for PiAdV-2. The standard equations are as follows (Fig 1): PiAdV-1:$Y = -2.815X + 37.073$, $R^2 = 0.998$, PiAdV-2:$Y = -3.570X + 38.625$, $R^2 = 0.997$. The results demonstrate the high efficiency and reliability of the newly developed dual real-time PCR assays.

### Specificity, sensitivity and repeatability

The amplification curves showed that only corresponding FAM and CY5 signals for PiAdV-1 and PiAdV-2, and negative for the AIV, RVA, NDV, PiCV, PiHV and ddH$_2$O (Fig 2). The above results indicated that the established assay had high specificity.

The sensitivity of the dual real-time PCR assay was assessed using 10-fold serial dilutions of standard plasmids, with concentrations ranging from $9.48 \times 10^5$ to $9.48 \times 10^0$ copies/μL for PiAdV-1 and from $8.84 \times 10^5$ to $8.84 \times 10^0$ copies/μL for

**Table 3. Adenoviruses representative strain information.**

| Strain name | Nation | Host | Year | GenBank |
|---|---|---|---|---|
| PAV/FJ2017 | China | Pigeon | 2017 | MF576429 |
| M144 | Hungary | Pigeon | 2011 | KX673408 |
| AH712 | China | Chicken | 2016 | KY436522 |
| CH/AHBZ/2015 | China | Chicken | 2015 | KU569295 |
| GDMZ | China | Chicken | 2016 | MG856954 |
| AG234-CORR | Mexico | Chicken | 1995 | MK572849 |
| 08-17832 | French | Chicken | 2008 | MK572857 |
| TNI1 | --- | Turkey | --- | NC022612 |
| 340 | Irish | Chicken | 1970 | KC493646 |
| JM1/1 | Japan | Chicken | 2000 | MF168407 |
| FAdV-1 | China | Fowl | 2017 | MK050972 |
| 1277BT | --- | Turkey | --- | NC022613 |
| P29 | Hungary | Goose | --- | JF510462 |
| P29 | Hungary | Goose | --- | NC017979 |
| AHAQ13 | China | Duck | 2017 | MH777396 |
| PL-pigeon-63/2023 | Polish | Pigeon | 2023 | PP999622 |
| PL-pigeon-4a/2023 | Polish | Pigeon | 2023 | PP999623 |
| CH-GD-12–2014 | China | Duck | 2014 | KR135164 |
| P18-05523–6 | Australia | Pigeon | 2018 | MW286325 |

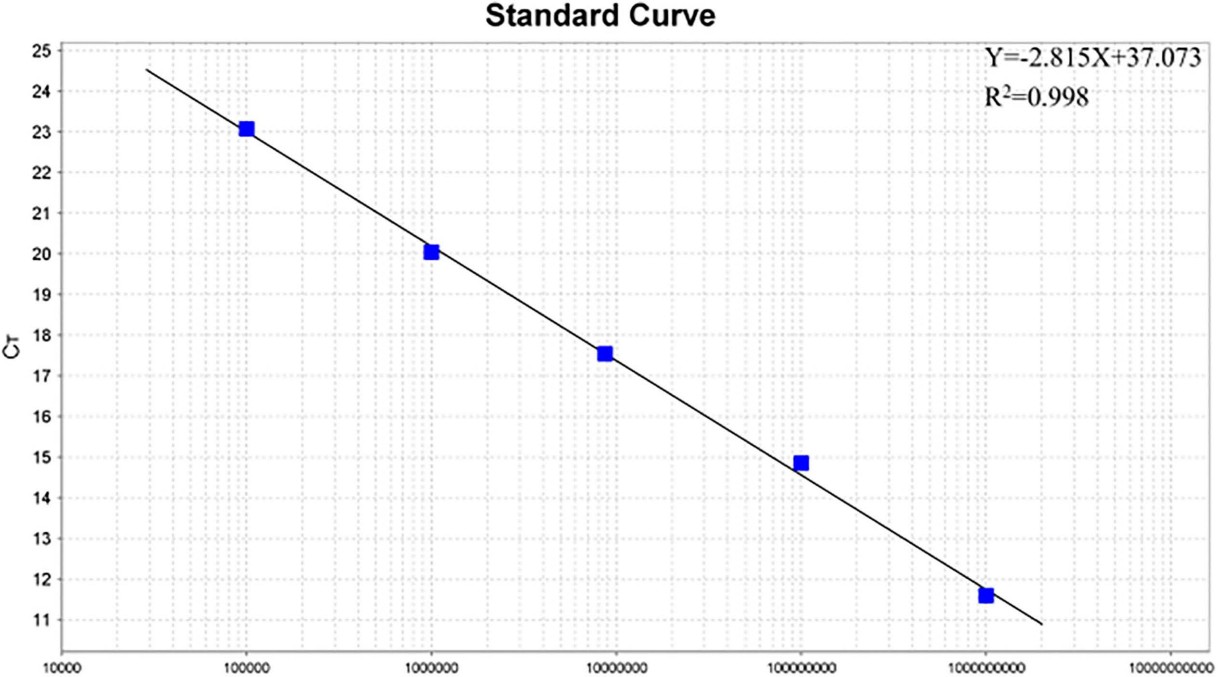

Target 2

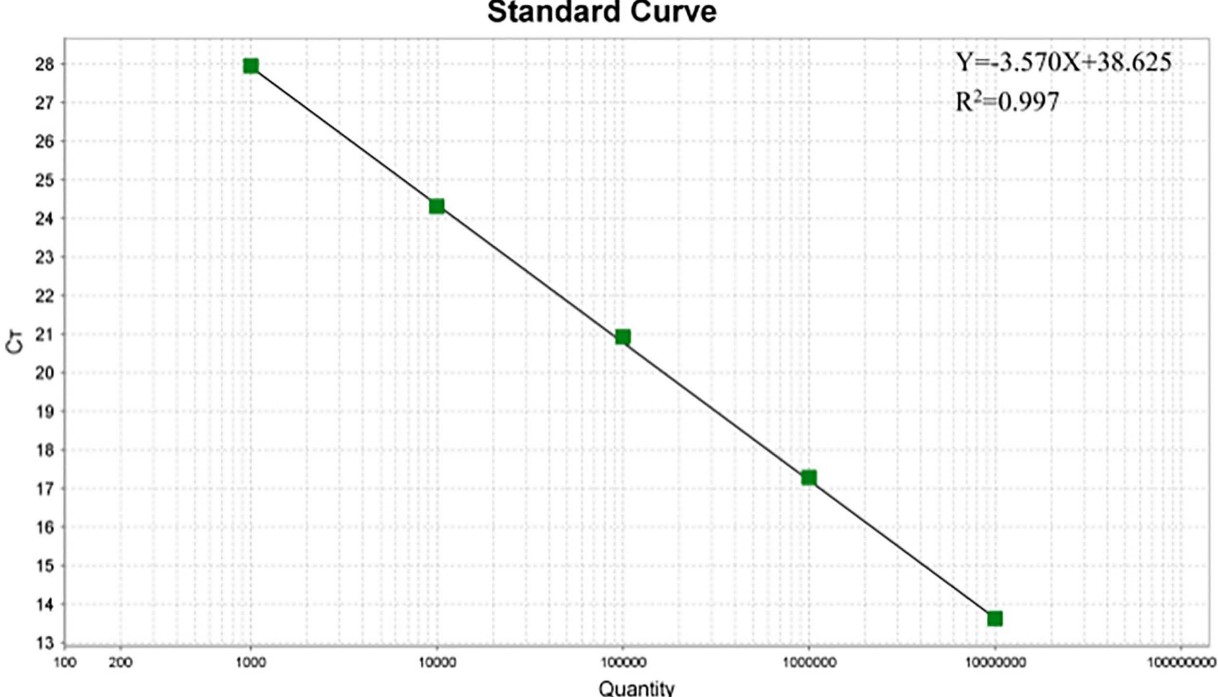

Target 1

**Fig 1. PiAdV-1 and PiAdV-2 dual real-time PCR detection standard curve.**

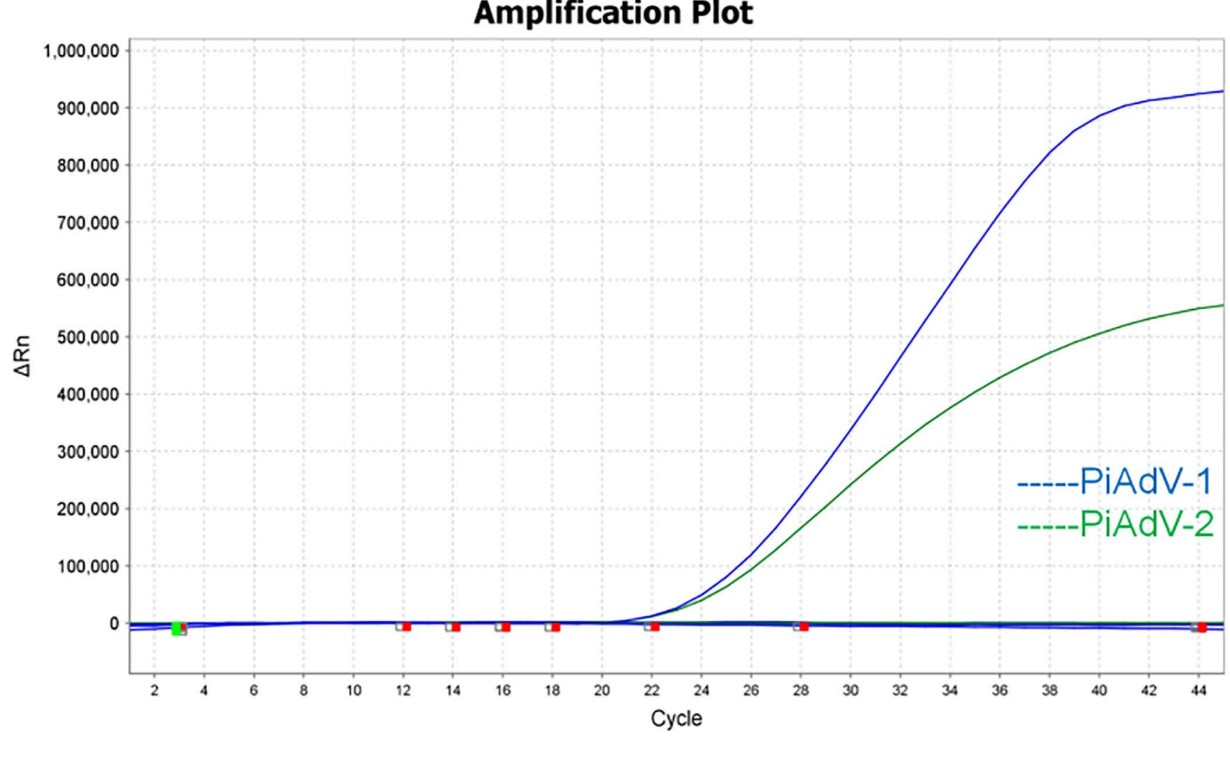

**Fig 2. Specificity test of the dual real-time PCR assay.**

PiAdV-2. Distilled water (ddH$_2$O) served as the negative control. The results indicated that the LOD for the assay was $9.48 \times 10^1$ copies/μL for PiAdV-1 and $8.84 \times 10^1$ copies/μL for PiAdV-2. (Fig 3).

The repeatability and reproducibility of the dual real-time PCR assay were evaluated using different concentrations of the corresponding standard plasmids. The intra-assay coefficient of variation (CV) was less than 0.02%, and the inter-assay CV was less than 0.03%, demonstrating that the dual real-time PCR assay offers high repeatability and reproducibility (Table 4).

### Clinical sample detection

The PiAdV-1 and PiAdV-2 dual real-time PCR method developed in this study was used to detect 20 samples with clinical suspicion of pigeon adenovirus infection. The results showed that two PiAdV-1 positive samples (10%), and thirteen PiAdV-2 positive samples (65%) were detected, followed by sequencing confirmed that 15 positive products were all pigeon adenovirus gene fragments. However, only one positive sample of PiAdV-1 and eight positive samples of PiAdV-2 were detected by conventional PCR using Table 2 primers (Table 5). The concordance rate between the two methods was 70.00%, with a Kappa value of 0.67.

### Epidemiological surveillance of PiAdV-1 and PiAdV-2 in China from 2022 to 2023

Oropharyngeal and cloacal swabs (n = 500) were collected from symptomless pigeons at LBMs in 10 provinces of China from 2022 to 2023. A total of 135 Pigeon adenovirus positive samples were detected by the dual real-time PCR assay

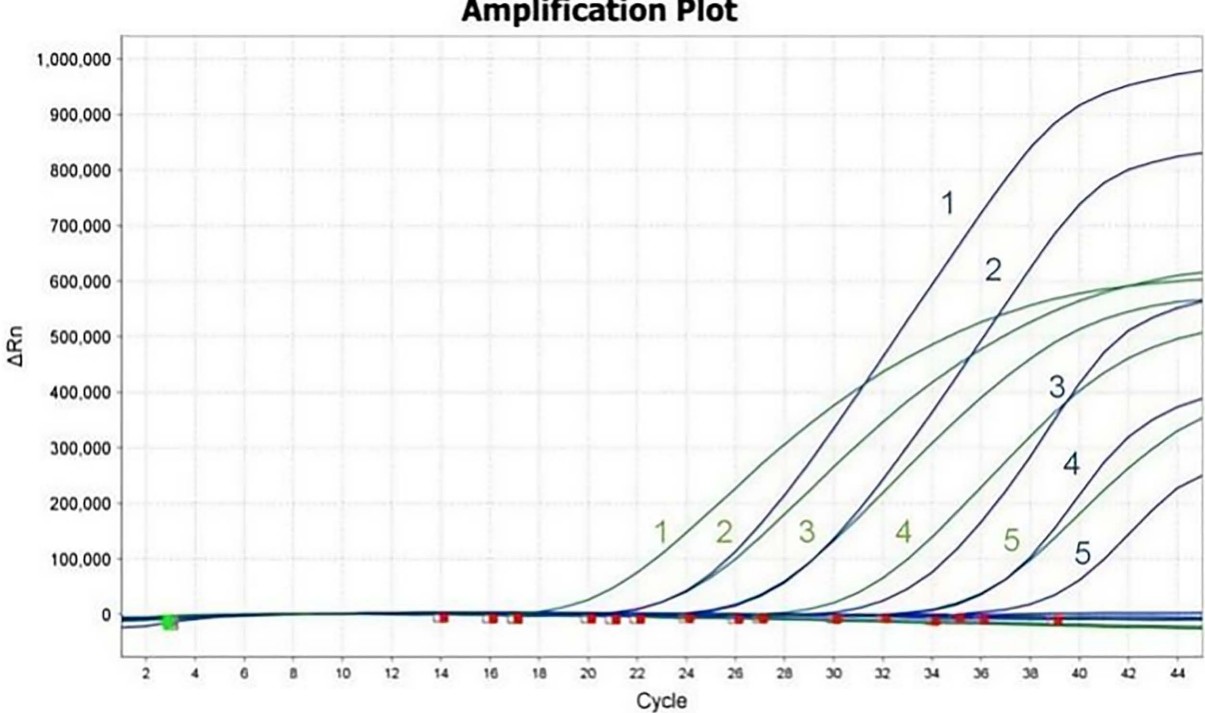

**Target 1** ■ **Target 2**

**Fig 3. Sensitivity test of the dual real-time PCR assay.** Labels 0–5 indicate different concentrations of plasmids, ranging from $9.48 \times 10^5$ to $9.48 \times 10^0$ copies/µL for PiAdV-1 and from $8.84 \times 10^5$ to $8.84 \times 10^0$ copies/µL for PiAdV-2, respectively.

**Table 4. Intra-repeatability and inter-reproducibility of the dual real-time PCR assay.**

| Virus | Standard (µL) | CT | Intra-Reproductivity | | | CT | Inter-Reproductivity | | |
|---|---|---|---|---|---|---|---|---|---|
| | | | Mean | SD | CV/% | | Mean | SD | CV/% |
| **PiAdV-1** | $1 \times 10^9$ | 11.153 | 11.043 | 0.10 | 0.01 | 11.588 | 11.161 | 0.37 | 0.03 |
| | | 10.979 | | | | 10.892 | | | |
| | | 10.998 | | | | 11.002 | | | |
| | $1 \times 10^8$ | 13.625 | 13.621 | 0.08 | 0.01 | 12.983 | 13.168 | 0.19 | 0.01 |
| | | 13.544 | | | | 13.154 | | | |
| | | 13.694 | | | | 13.368 | | | |
| | $1 \times 10^7$ | 16.143 | 16.164 | 0.09 | 0.01 | 16.089 | 16.176 | 0.08 | 0.00 |
| | | 16.084 | | | | 16.237 | | | |
| | | 16.265 | | | | 16.203 | | | |
| **PiAdV-2** | $1 \times 10^6$ | 17.301 | 17.330 | 0.07 | 0.00 | 17.529 | 17.532 | 0.06 | 0.00 |
| | | 17.279 | | | | 17.477 | | | |
| | | 17.411 | | | | 17.589 | | | |
| | $1 \times 10^5$ | 21.101 | 21.007 | 0.09 | 0.00 | 21.225 | 21.210 | 0.10 | 0.00 |
| | | 20.994 | | | | 21.301 | | | |
| | | 20.927 | | | | 21.103 | | | |
| | $1 \times 10^4$ | 23.570 | 24.122 | 0.49 | 0.02 | 23.984 | 24.205 | 0.24 | 0.01 |
| | | 24.493 | | | | 24.177 | | | |
| | | 24.303 | | | | 24.454 | | | |

**Table 5. PiAdV-1 and PiAdV-2 dual real-time fluorescence PCR vs conventional PCR assay comparison of results.**

| Clinical sample | Dual PCR | Conventional PCR |
|---|---|---|
| | Positive sample (rate) | Positive sample (rate) |
| PiAdV-1 | 2(10%) | 1(5%) |
| PiAdV-2 | 13(65%) | 8(40%) |

in the 10 surveyed provinces of Hunan, Henan, Jiangsu, Sichuan, Anhui, Guangxi, Guizhou, Hubei, Guangdong, and Jiangxi. The total positive rate of Pigeon adenovirus was 27.00% (135/500) (Table 6). The positivity rates of PiAdV-1 and PiAdV-2 were 2.4% (12/500) and 25.4% (127/500), respectively. There were 8 positive samples with single PiAdV-1 (1.6%), 123 positive samples with single PiAdV-2 (24.6%), and 4 positive samples with PiAdV-1 and PiAdV-2 together (0.8%). The positive rate in Guizhou was the highest (51.43%) and the positive rate in Anhui was the lowest (8.51%).

## Sequencing and phylogenetic analysis

Fifteen representative viral strains from different regions were sequenced, and the phylogenetic tree of pigeon adenovirus Hexon gene was analysed. The results of phylogenetic analysis showed that 10 samples belonged to the PiAdV-2 subtype and 5 samples belonged to the PiAdV-1 subtype (Fig 4). The pigeon adenovirus positive samples of C2182, A2023, J2477, E2459, Q2428, X2236 and K2457 in this study all belong to the variant B subbranch of PiAdV-2, which is the closest relative to the PAV/F2017 [11] strain detected in China in 2018, with nucleotide homology ranging from 99.5% to 99.7%. The G2567, S2569 and H2021 belonged to the variant A subbranch of PiAdV-2, and were closely related to YPDS-Y-V1 and M144 strains detected in China, with nucleotide homology ranging from 87.6% to 99.7%. The positive samples of H2374, X2296, A2452, S2227 and E2334 all belonged to the PiAdV-1, and were closely related to P18-05523–6 strains detected in 2021, with nucleotide homology of 99.5%.

## Discussion

In recent years, the pigeon raising industry in China has gradually developed in industrialization and scale. The number of pigeons raised has increased year by year in China, and people have paid more and more attention to the disease in pigeon flocks. PiAdV-1 and PiAdV-2 are common pathogens in pigeons. The pigeons infected with PiAdV-1 and PiAdV-2 mainly show vomiting and digestive stagnation [13], followed by dehydration, temperature rise, weight loss, green water

**Table 6. Positive annual rates of Pigeon adenovirus detection in 2022-2023.**

| Provinces | number of samples | single PiAdV-1 | single PiAdV-2 | PiAdV-1+PiAdV-2 | Positive rate (%) (95%CI) |
|---|---|---|---|---|---|
| Guangdong | 40 | 0 | 9 | 0 | 22.50(9.56-35.44) |
| Hubei | 47 | 1 | 4 | 0 | 8.51(0.53-16.49) |
| Jiangxi | 70 | 1 | 27 | 2 | 41.43(29.89-52.97) |
| Anhui | 48 | 2 | 2 | 0 | 8.33(0.51-16.15) |
| Guangxi | 70 | 3 | 14 | 1 | 25.71(15.47-35.95) |
| Guizhou | 35 | 0 | 18 | 0 | 51.43(34.87-67.99) |
| Jiangsu | 79 | 0 | 12 | 0 | 15.19(7.28-23.10) |
| Sichuan | 15 | 0 | 2 | 0 | 13.33(3.87-30.53) |
| Hunan | 28 | 0 | 9 | 0 | 32.14(14.84-49.44) |
| Henan | 68 | 1 | 26 | 1 | 41.18(29.48-52.88) |
| Total | 500 | 8 | 123 | 4 | 26.60(22.73-30.47) |

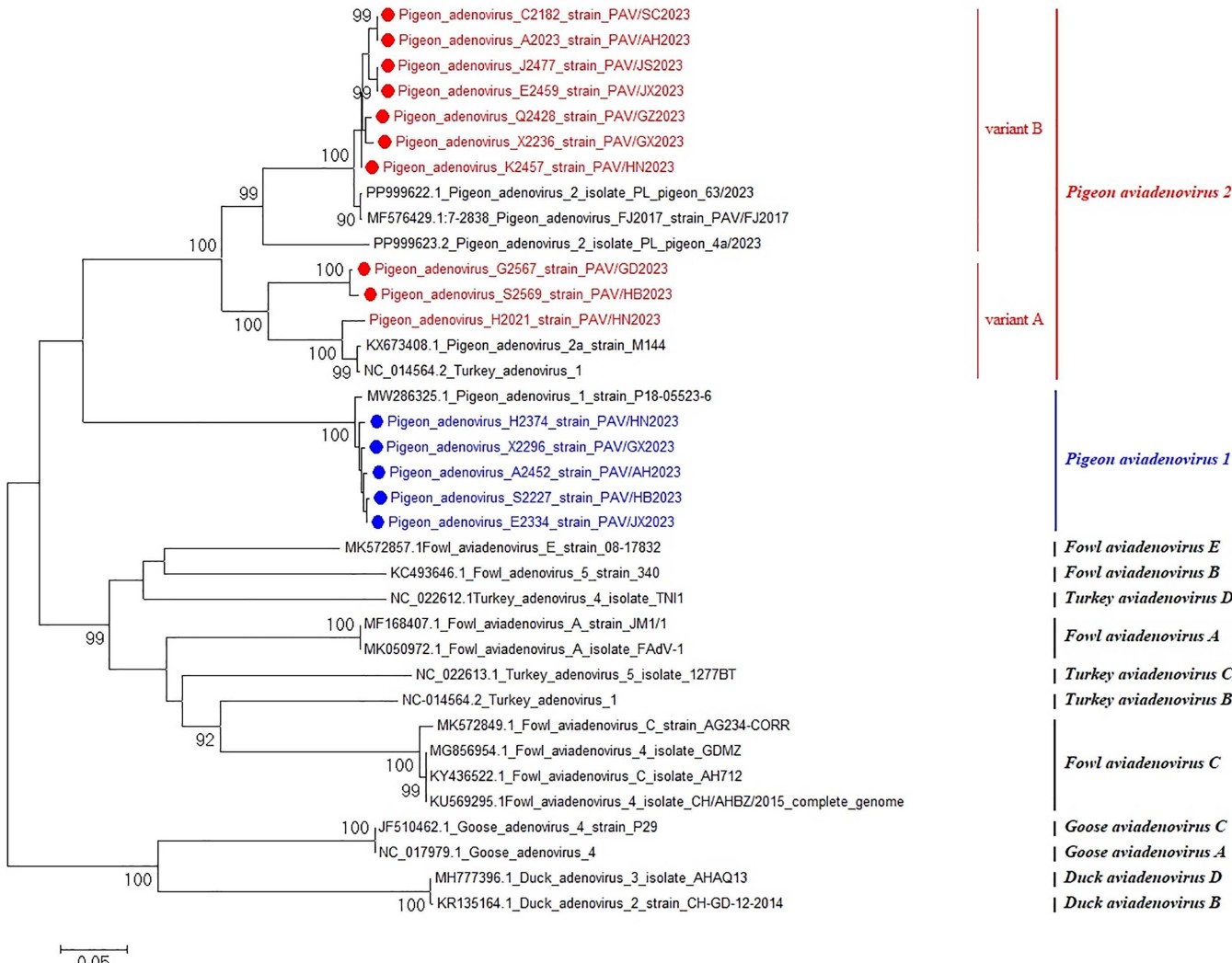

**Fig 4. The phylogenetic tree was constructed based on the nucleotide sequences of the PiAdVs Hexon gene.** Samples indicated by "●●" represent the positive samples tested in this study.

excretion or other phenomena [14,15]. The effects of PiAdV-1 infection on pigeons are strikingly similar to YPDS in that it primarily affects young pigeons and manifests with diarrhea, vomiting and weight loss for about a week [9,11]. PiAdV-2 affects pigeons of all ages and is characterized by sudden death and extensive liver lesions. Immunocompromised pigeons inoculated with dexamethasone treated liver homogenate supernatant of natural infection cases could reproduce the disease [7]. When PiAdV-1 or PiAdV-2 infected pigeons alone, the mortality rate was low. However when PiAdV-1 or PiAdV-2 infected pigeons with other pathogens, the mortality rate of pigeons increased significantly, which caused serious economic losses to the pigeon industry. Kalpana [16] reported a case of sudden death in 28 pigeons in 2018, which showed that all 28 pigeons died from a combination of PiAdV-1 and pigeon torque teno virus (PTTV). Chen et al [4] showed that the prevalence of PiAdV-2 was 37.14% in YPDS racing pigeons and 20.93% in healthy racing pigeons.

Early diagnosis of pathogens is a key factor in prevention and control of PiAdV [17]. In 2002, Raue et al.[12] developed a conventional PCR using the PiAdV-1 fiber. In 2024, Chen et al. [4] developed a TaqMan-qPCR assay using the PiAdV-2

fiber-2 gene by designing specific primers and probes. Łukaszuk et al.[1] developed a TaqMan qPCR assay targeting a 135 bp fragment of the PiAdV-2 protease-coding gene. At present, there are only separate nucleic acid detection methods based on PiAdV-1 and PiAdV-2. Therefore, it is urgent to establish a sensitive and specific high-throughput rapid detection method for PiAdV-1 and PiAdV-2, providing technical reserves for the rapid clinical diagnosis of PiAdV-1 and PiAdV-2 [11], to achieve early detection, early isolation and early treatment of pigeon flock, thereby reducing the loss of pigeon flocks. In this study, specific primers and probes were designed based on the conserved Hexon gene sequences of PiAdV-1 and PiAdV-2 in pigeons, and the reaction system and reaction conditions were optimized to establish a dual real-time fluorescent PCR method for detection of PiAdV-1 and PiAdV-2. The detection method has strong specificity, high sensitivity and good repeatability. Only the nucleic acid of PiAdV-1 and PiAdV-2 showed a specific amplification curve. The lowest detection limits were 94.8 copies/µL and 88.4 copies/µL, respectively. The coefficient of variation of PiAdV-1 and PiAdV-2 was less than 0.03%, which showed good repeatability. The pigeon adenovirus dual real-time fluorescent PCR method established in this study and the conventional PCR method were used to detect the etiology of 20 clinically suspected PiAdV infection samples. The results showed that two PiAdV-1 positive samples and thirteen PiAdV-2 positive sample were detected. Meanwhile, all positive samples were confirmed by sequencing. However, only one positive sample of PiAdV-1 and eight positive samples of PiAdV-2 were detected by conventional PCR. It is suggested that the pigeon adenovirus dual real-time fluorescent PCR method established in this study has higher sensitivity and good clinical diagnostic performance than the conventional PCR method. In summary, this study established a rapid, sensitive, specific and accurate dual real-time fluorescent PCR detection method for PiAdV-1 and PiAdV-2, providing necessary technical support for the early diagnosis and comprehensive prevention and control of PiAdV-1 and PiAdV-2.

In order to understand the epidemiologic state of PiAdV-1 and PiAdV-2 in pigeon flocks in China, a total of 500 oro-pharyngeal and cloacal swabs were collected from LBMs in 10 provinces in China from 2022 to 2023, and the dual fluorescent PCR established in this study was used to detect PiAdV-1 and PiAdV-2. The result showed that 135 positive samples were detected, in surveyed 10 provinces with a positive rate of 27.00%, and the highest positive rate was 51.43% in Guangzhou Province. Overall, PiAdvs showed a relatively high epidemic trend in China. PiAdV-1 single infection was detected in 8 cases (1.6%), PiAdV-2 single infection in 123 cases (24.6%), and PiAdV-1 and PiAdV-2 co-infection was detected in 4 cases (0.8%), which showed that the PiAdV-2 was the dominant virus in Chinese pigeons. At the same time, there was a phenomenon of PiAdV-1 and PiAdV-2 co-infection in the pigeons, but the infection rate is not high. In the clinical detection of diseased pigeon samples in our lab, we also detected PiAdV-1 single infections, PiAdV-2 single infections, and mixed infections of PiAdV-1 and PiAdV-2. The clinical detect results showed that PiAdV-2 was detected more frequently than PiAdV-1, which is basically consistent with the results of our epidemiological investigation. Overall, the prevalence of PiAdV-2 is much higher than that of PiAdV-1 and it is the dominant strain in pigeon flocks in China. While we were unsuccessful in isolating the virus, we cannot conduct animal regression tests to determine their pathogenicity. In 1993−1994, Herdt et al.[7] found that 151 pigeons out of 988 were infected with PiAdV-2 among the Belgian pigeon population. Mónika [18] conducted a 4-year survey of healthy and sick pigeons in 27 lofts in Hungary and showed that the pigeon adenovirus positivity rate was as high as 50%. These indicated that pigeon adenovirus was widespread in pigeons all over the world.

For 15 representative positive virus strains from different provinces detected by dual real-time PCR that were sequenced by using the primers specific. The phylogenetic analysis revealed that 10 strains were PiAdV-2 and 5 strains were PiAdV-1. Among the 10 PiAdV-2 strains, the homology of 7 strains with the Chinese PAV/F2017 strain was 99.5% − 99.7%, forming conformed clusters (variation B), suggesting the existence of epidemiological associations. The other three strains of PiAdV-2 (variant A) have a genetic diversity of 87.6–99.7% compared with the domestic reference sequence, indicating that there are variations in their evolution. All five PiAdV-1 strains and the reference sequence showed high conservation (99.5%), which might reflect a slower evolution. These findings demonstrate the genetic diversity of PiAdV in pigeon flocks in our country, which also indicates the difficulty in the prevention and control of PiAdV in China.

In summary, a dual real-time PCR assay for differential detection of PiAdV-1 and PiAdV-2 was successfully established in this study, which showed good specificity, sensitivity, repeatability, and feasibility. In addition, the prevalence of PiAdV-1 and PiAdV-2 infection in pigeon flocks in China was first evaluated using this established method, and the results showed that pigeon adenovirus was widely distributed and has a high positive rate in pigeon flocks in China. Therefore, continued surveillance studies and the PiAdV-1 and PiAdV-2 vaccine development should be carried out to control PiAdV-1 and PiAdV-2 infection in pigeons in China.

## Acknowledgments

This study was funded by the National Key Research & Development Program (2022YFD1800600).The funders had no role in study design, data collection and analysis, decision to publish, or preparation of the manuscript.

## Author contributions

**Data curation:** Yufeng Liu, Haiming Wang, Xiaohui Yu.

**Formal analysis:** Jinping Li.

**Funding acquisition:** Wenming Jiang, Hualei Liu.

**Investigation:** Lu Chen, Yuteng Chen, Wenming Jiang.

**Methodology:** Xiaohui Yu.

**Resources:** Hualei Liu, Xiaohui Yu.

**Visualization:** Jingjing Wang.

**Writing – original draft:** Lu Chen.

**Writing – review & editing:** Lu Chen.

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
