## [Decision Letter · Decision Letter 0]

29 Jan 2025

Dear Dr. Chen,

Thank you for submitting your manuscript to PLOS ONE. After careful consideration, we feel that it has merit but does not fully meet PLOS ONE’s publication criteria as it currently stands. Therefore, we invite you to submit a revised version of the manuscript that addresses the points raised during the review process.

**ACADEMIC EDITOR:**

Dear Dr Chen,

Your manuscript, "Establishment and application of a dual real-time PCR assay for differential detection of PiAdV-A and PiAdV-B among pigeons in China in 2022-2023", has now been assessed.

We invite you to revise your paper, carefully addressing the comments from the reviewers.  Please ensure the results are accurately reported, any overstated conclusions are rewritten and the limitations of the work fully explained. When your revision is ready, please submit the updated manuscript and a point-by-point response. This will help us move to a swift decision.

We look forward to receiving your revised manuscript.

Kind regards,

Ahmed Eisa Elhag

Academic Editor

PLOS ONE

Journal Requirements:

2. Thank you for submitting your work to PLOS ONE. We note that you have not mentioned the source of tissue samples from pigeons used in this study. Before we can continue with your submission, kindly mention the details in the Method section. Thank you for your attention to this request. We look forward to hearing from you.

https://www.mdpi.com/1999-4915/15/6/1238

https://onlinelibrary.wiley.com/doi/10.1111/tbed.14464

In your revision ensure you cite all your sources (including your own works), and quote or rephrase any duplicated text outside the methods section. Further consideration is dependent on these concerns being addressed.

6. We note that Figure 4 in your submission contain [map/satellite] images which may be copyrighted. All PLOS content is published under the Creative Commons Attribution License (CC BY 4.0), which means that the manuscript, images, and Supporting Information files will be freely available online, and any third party is permitted to access, download, copy, distribute, and use these materials in any way, even commercially, with proper attribution. For these reasons, we cannot publish previously copyrighted maps or satellite images created using proprietary data, such as Google software (Google Maps, Street View, and Earth). For more information, see our copyright guidelines: http://journals.plos.org/plosone/s/licenses-and-copyright.

a. You may seek permission from the original copyright holder of Figure 4 to publish the content specifically under the CC BY 4.0 license.  

**Reviewers' comments:**

Reviewer's Responses to Questions

**Comments to the Author**

1. Is the manuscript technically sound, and do the data support the conclusions?

Reviewer #1: Partly

Reviewer #2: Partly

2. Has the statistical analysis been performed appropriately and rigorously?

Reviewer #1: N/A

Reviewer #2: N/A

3. Have the authors made all data underlying the findings in their manuscript fully available?

Reviewer #1: Yes

Reviewer #2: Yes

4. Is the manuscript presented in an intelligible fashion and written in standard English?

Reviewer #1: Yes

Reviewer #2: Yes

**Reviewer #1** : 1. The ICTV information provided for Adenovirus in the introduction is not up to date. It would be appropriate to change this information to its current state.

2. The distinction between RNA and DNA viruses presented under the heading "2.5 Specificity and Sensitivity" was made incorrectly. In addition, the same abbreviation was used for both coronavirus and circovirus. It would be appropriate to correct these errors.

3. The data presented in Table 3 and Table 5 are a bit confusing. (Example: Do PiAdV-A and PiAdV-B have the same virus dilution copies? It would be more appropriate to write the positive rate separately in Table 5.)

**Reviewer #2** : This manuscript aimed to validate a reliable analytical method for direct and dual detection of PiAdV-A and PiAdV -B genome by real-time PCR in pigeon samples. The authors claim that their validated PCR assay provide accurate and reliable screening to detect and differentiate PiAdV-A and PiAdV-B, and provide valuable information on the prevalence of PiAdV in China. They report that their method could be a useful tool in practice for detection of PiAdV and to provide information on epidemiologic surveillance of PiAdV. Indeed, in the literature, few studies are available on PiAdV detection methods and little is known about PiAdV prevalence in China. The study provides a detection method for PiAdV genome direct detection by PCR, even though the LOD should be determined more precisely. The study also provide data about PiAdV prevalence in China. A number of clarifications are required.

Major comments:

The exact usefulness of the differentiation of PiAdV-A and B is not clear in the introduction and the discussion. Aside from the epidemiological surveillance, how useful is the dual detection in routine farm diagnosis? Line 64-65, the authors mentioned that distinction between PiAdV-A and B is mandatory for “rapid clinical diagnosis”. Why the detection of total PiAdV is not sufficient to contain epidemics in breeding farms and what will the dual detection change in the management of epidemics in a breeding farm? The PiAdV-A prevalence appears to be very low compare to type B. Why is it relevant to differentiate between the two types in routine diagnosis?

The authors should develop these points in the introduction and discussion.

Most papers about PiAdV talk about PiAdV type 1 or 2 and not variant A and B (Teske et al. 2017; Chen et al. 2024).

Line 36. Ref 2 about “Falcon adenovirus B” refer to a paper that identified PiAdV type 2, with variant A and B that belong to the PiAdV-2. This is very confusing. The authors speak about PiAdV-1 and 2 or A and B belonging to PiAdV-2? In the phylogenetic tree of the present study (figure 5), types identified are 1 and 2 and not A and B. This is very confusing. Please revised the nomenclature.

The comments below retain the original A and B nomenclature of the manuscript under revision.

Lines 52-54. Confusing. How it can be clear that the disease has been prevalent in China? Line 63, the authors mentioned that the prevalence is not clear in China. If the prevalence is not known, so are the economic losses, and studies are needed to assess prevalence in China and the associated economic losses. Please revise this point.

Lines 55-59. The authors mentioned that the virus isolation is the more reliable detection method. Is it a general comment or specific to PiAdV detection? Has virus isolation been used for PiAdV diagnosis? Is there some studies showing better PiAdV detection with virus isolation than PCR? Real-time PCR is usually highly sensitive.

Lines 57-60. Serum neutralizing antibody technology is not a common method for viral detection in laboratory, but useful for specific studies about protection of an individual or a population against a virus (seroprevalence). Seroprevalence of PiAdV neutralizing antibodies could be an interesting tool to determine the circulation of PiAdV in pigeons and for epidemiological surveillance. The added-value of seroprevalence and serum neutralizing antibody technology may be broach in the discussion and not the introduction.

Line 121. Please specify if testing for a single dilution was performed from one tube or from three different samples (for example three different samples of 9.48x10^9 copies of PiAdV-A). The guidelines from World Organisation for Animal Health (WOAH - “PRINCIPLES AND METHODS OF VALIDATION OF DIAGNOSTIC ASSAYS FOR INFECTIOUS DISEASES” p11) specify that “It is not acceptable to prepare a final working dilution of a sample in a single tube from which diluted aliquots are pipetted into reaction vessels, or to create replicates from one extraction of nucleic acid rather than to extract each replicate before dilution into the reaction vessels. Such ‘samples’ do not constitute valid replicates for repeatability studies”.

Line 131. About the conventional PCR. Reference 12 refer to a panadenovirus nested-PCR, and reference 13 to a PCR detecting specifically total PiAdV. Have these two methods been performed and shown the same results? Please clarify which conventional PCR has been performed in the present study. Specify the LOD of the conventional PCR assay if known.

Table 4. Indicate more clearly in the table which one is the conventional PCR and which one is the dual.

Line 147. How many negative sample (ddH2O) were tested?

Lines 153-156. How many times each point of the 10-fold serially diluted standard plasmids have been tested to determine the LOD for PiAdV-A and B genome? It is frequent to see 3 to 5 testing for each dilution of the standard, to ensure the 100% positive detection for the last point. A single test for each dilution is not robust enough. Did the authors test additional dilution beyond 94.8 or 88.4 copies/µL and 0 copy /µL?

“With a 10-fold serially diluted standard plasmids, the last dilution showing 100% response may be accepted as a conservative estimate of the lower limit of detection. A more accurate estimate may be obtained by a second stage experiment using narrower intervals in the dilution scheme focusing on the region between 100% and 0%” (i.e. in this study, between 94.8 or 88.4 and 0 copy/µL in this study) (ref WOAH- “PRINCIPLES AND METHODS OF VALIDATION OF DIAGNOSTIC ASSAYS FOR INFECTIOUS DISEASES”)

Discussion.

The authors claim that the method developed for dual detection is satisfactory, but the limit of detection is higher than the LOD of 34 copies /µL for PiAdV-2 PCR developed by Chen et al. 2024. This point should be discussed, in addition to the added value of the dual real-time PCR developed by the authors in comparison with the results of Chen et al. 2024.

Line 194-196. The authors mentioned that the co-infection between PiAdV and other pathogens increase the mortality rate of pigeons. Information about co-infection with PTTV appear later, line 231. This is confusing. Is there a point to discuss about co-infection with other pathogens and PiADV-A and –B infection?

Could the type of PiAdV affect the mortality rate during such co-infection? Is it important to develop a multiplex PCR detecting those pathogens and PiAdV rather than differentiating PiAdV-A and B?

The authors should clarify their point about co-infection with other pathogens and their results.

Minor comments:

The authors mentioned a limit of detection of 10^1 copies/µL for PiAdV-A and B in the abstract, but 94.8 copies for PiAdV-A and 88.4 for PiAdV-B are mentioned in the results and discussion. Indicate the exact LOD for each PiAdV type in the abstract and homogenise in the results. No need for 101, the entire number is more comprehensible.

Line 55. Can the authors specify if the diagnosis of total PiAdV by PCR is routinely used in breeding farms, in case of pigeon diarrhoea or death?

Line 92. Please specify what type of tissue samples (n=20) from pigeons were tested.

Lines 220-221. Is it known if a high morbidity or mortality rate of pigeons in the breeding farm of the regions with the higher PiAdV prevalence was observed?

Improve English phrasing: e.g.:

Line 23. The dual fluorescence PCR method established in this study was used to test 500 pigeon swab samples.

Line 37. The virion has icosahedral symmetry […] the capsid proteins mainly consist of […].

Line 56. “has long detection cycle”, did the authors mean “is time-consuming” ?

Lines 58-59. but these need to the more higher viral load of the sample. This need to be reworded.

Line 138. Delete parenthesis

Line 195. Correct pegions.

References

Chen C, Zhu C, Chen Z, Cai G, Lin L, Zhang S, Jiang B, Miao Z, Fu G, Huang Y, Wan C. Rapid detection of pigeon adenovirus 2 using a TaqMan real-time PCR assay. Poult Sci. 2024 Jul;103(7):103848. doi: 10.1016/j.psj.2024.103848. Epub 2024 May 16. PMID: 38843610; PMCID: PMC11216009.

Teske L, Rubbenstroth D, Meixner M, Liere K, Bartels H, Rautenschlein S. Identification of a novel aviadenovirus, designated pigeon adenovirus 2 in domestic pigeons (Columba livia). Virus Res. 2017 Jan 2;227:15-22. doi: 10.1016/j.virusres.2016.09.024. Epub 2016 Sep 30. PMID: 27697452.

**Do you want your identity to be public for this peer review?** For information about this choice, including consent withdrawal, please see our Privacy Policy

Reviewer #1: **Yes: ** HARUN ALBAYRAK

Reviewer #2: No

---

## [Author Response · Author response to Decision Letter 1]

3 Apr 2025

To Journal Requirements:

1. Please ensure that your manuscript meets PLOS ONE's style requirements, including those for file naming. The PLOS ONE style templates can be found at https://journals.plos.org/plosone/s/file?id=wjVg/PLOSOne_formatting_sample_main_body.pdf and https://journals.plos.org/plosone/s/file?id=ba62/PLOSOne_formatting_sample_title_authors_affiliations.pdf.

Answer: The paper format and submission documents have been revised and submitted in accordance with the journal's requirements.

2.Thank you for submitting your work to PLOS ONE. We note that you have not mentioned the source of tissue samples from pigeons used in this study. Before we can continue with your submission, kindly mention the details in the Method section. Thank you for your attention to this request. We look forward to hearing from you.

Answer: The source information of pigeon tissue samples has been added in the manuscript

3.We noticed you have some minor occurrence of overlapping text with the following previous publication(s), which needs to be addressed: https://www.mdpi.com/1999-4915/15/6/1238
https://onlinelibrary.wiley.com/doi/10.1111/tbed.14464.

In your revision ensure you cite all your sources (including your own works), and quote or rephrase any duplicated text outside the methods section. Further consideration is dependent on these concerns being addressed.

Answer: We cited relevant literature and rephrased duplicate texts in the manuscript

4.We note that the grant information you provided in the ‘Funding Information’ and ‘Financial Disclosure’ sections do not match.When you resubmit, please ensure that you provide the correct grant numbers for the awards you received for your study in the ‘Funding Information’ section.

Answer: The information in the funding section was confirmed.

5.Your ethics statement should only appear in the Methods section of your manuscript. If your ethics statement is written in any section besides the Methods, please delete it from any other section.

Answer: We have removed the other parts of the ethics statement.

6.We note that Figure 4 in your submission contain [map/satellite] images which may be copyrighted. All PLOS content is published under the Creative Commons Attribution License (CC BY 4.0), which means that the manuscript, images, and Supporting Information files will be freely available online, and any third party is permitted to access, download, copy, distribute, and use these materials in any way, even commercially, with proper attribution. For these reasons, we cannot publish previously copyrighted maps or satellite images created using proprietary data, such as Google software (Google Maps, Street View, and Earth).

Answer: Figure 4 has been removed and the relevant content was presented as text in the manuscript.

To Reviewer #1:

1.The ICTV information provided for Adenovirus in the introduction is not up to date. It would be appropriate to change this information to its current state.

Answer: Thanks for your comment. The relevant content has been modified according to the latest ICTV adenovirus information in the manuscript.

2. The distinction between RNA and DNA viruses presented under the heading "2.5 Specificity and Sensitivity" was made incorrectly. In addition, the same abbreviation was used for both coronavirus and circovirus. It would be appropriate to correct these errors.

Answer: Thanks for your advice. The distinction between RNA and DNA viruses presented have been modified. The abbreviations for coronaviruses and circoviruses have been amended in the manuscript

3. The data presented in Table 3 and Table 5 are a bit confusing. (Example: Do PiAdV-A and PiAdV-B have the same virus dilution copies? It would be more appropriate to write the positive rate separately in Table 5.)

The author,s answer: Thanks for your suggestion, and the contents of Tables 3 and 5 have been modified as suggested.

To Reviewer #2:

1.The exact usefulness of the differentiation of PiAdV-A and B is not clear in the introduction and the discussion. Aside from the epidemiological surveillance, how useful is the dual detection in routine farm diagnosis? Line 64-65, the authors mentioned that distinction between PiAdV-A and B is mandatory for “rapid clinical diagnosis”. Why the detection of total PiAdV is not sufficient to contain epidemics in breeding farms and what will the dual detection change in the management of epidemics in a breeding farm? The PiAdV-A prevalence appears to be very low compare to type B. Why is it relevant to differentiate between the two types in routine diagnosis?

The authors should develop these points in the introduction and discussion.

The author,s answer: Thanks for your comment. The explanation of the need to distinguish between PIADVA and B has been further elaborated in the introduction and discussion.

2.Most papers about PiAdV talk about PiAdV type 1 or 2 and not variant A and B (Teske et al. 2017; Chen et al. 2024).

Line 36. Ref 2 about “Falcon adenovirus B” refer to a paper that identified PiAdV type 2, with variant A and B that belong to the PiAdV-2. This is very confusing. The authors speak about PiAdV-1 and 2 or A and B belonging to PiAdV-2? In the phylogenetic tree of the present study (figure 5), types identified are 1 and 2 and not A and B. This is very confusing. Please revised the nomenclature.

The author,s answer: We have changed the nomenclature of PiAdV-A and PiAdV-B as suggested.

3.Lines 52-54. Confusing. How it can be clear that the disease has been prevalent in China?

Line 63, the authors mentioned that the prevalence is not clear in China. If the prevalence is not known, so are the economic losses, and studies are needed to assess prevalence in China and the associated economic losses. Please revise this point.

The author,s answer: Thanks for your suggestion. The confusing description has been revised

4.Lines 55-59. The authors mentioned that the virus isolation is the more reliable detection method. Is it a general comment or specific to PiAdV detection? Has virus isolation been used for PiAdV diagnosis? Is there some studies showing better PiAdV detection with virus isolation than PCR? Real-time PCR is usually highly sensitive.

The author,s answer: Thanks for your comment. This description that the virus isolation is the more reliable detection method refers to a generic description of all viruses. At present, the isolation of pigeon adenovirus in our laboratory is not go well. For this puzzling, the relevant content has been redescribed in the manuscript.

5.Lines 57-60. Serum neutralizing antibody technology is not a common method for viral detection in laboratory, but useful for specific studies about protection of an individual or a population against a virus (seroprevalence). Seroprevalence of PiAdV neutralizing antibodies could be an interesting tool to determine the circulation of PiAdV in pigeons and for epidemiological surveillance. The added-value of seroprevalence and serum neutralizing antibody technology may be broach in the discussion and not the introduction.

The author,s answer:Thanks for your advice. Changes have been made as your suggestion.

6.Line 121. Please specify if testing for a single dilution was performed from one tube or from three different samples (for example three different samples of 9.48x10^9 copies of PiAdV-A). The guidelines from World Organisation for Animal Health (WOAH - “PRINCIPLES AND METHODS OF VALIDATION OF DIAGNOSTIC ASSAYS FOR INFECTIOUS DISEASES” p11) specify that “It is not acceptable to prepare a final working dilution of a sample in a single tube from which diluted aliquots are pipetted into reaction vessels, or to create replicates from one extraction of nucleic acid rather than to extract each replicate before dilution into the reaction vessels. Such ‘samples’ do not constitute valid replicates for repeatability studies”.

The author,s answer: Due to a mistake in my language description, which led to the reviewer's doubt, some of the language in Line 126-130 (formerly Line 121) has now been revised.

7.Line 131. About the conventional PCR. Reference 12 refer to a panadenovirus nested-PCR, and reference 13 to a PCR detecting specifically total PiAdV. Have these two methods been performed and shown the same results? Please clarify which conventional PCR has been performed in the present study. Specify the LOD of the conventional PCR assay if known.

Table 4. Indicate more clearly in the table which one is the conventional PCR and which one is the dual.

The author,s answer: Primers used for conventional PCR have been shown in Table 2 and more clearer description is added to Table 4.

8.Line 147. How many negative sample (ddH2O) were tested?

The author,s answer: The negative samples included 3 AIV, 1 RVA, 1 NDV, 1 PiCV, 1PiHV and 1 negative control ddH2O. The relevant content is redescribed in the manuscript.

9.Lines 153-156. How many times each point of the 10-fold serially diluted standard plasmids have been tested to determine the LOD for PiAdV-A and B genome? It is frequent to see 3 to 5 testing for each dilution of the standard, to ensure the 100% positive detection for the last point. A single test for each dilution is not robust enough. Did the authors test additional dilution beyond 94.8 or 88.4 copies/µL and 0 copy /µL?

“With a 10-fold serially diluted standard plasmids, the last dilution showing 100% response may be accepted as a conservative estimate of the lower limit of detection. A more accurate estimate may be obtained by a second stage experiment using narrower intervals in the dilution scheme focusing on the region between 100% and 0%” (i.e. in this study, between 94.8 or 88.4 and 0 copy/µL in this study) (ref WOAH- “PRINCIPLES AND METHODS OF VALIDATION OF DIAGNOSTIC ASSAYS FOR INFECTIOUS DISEASES”)

The author,s answer: Thanks for your comment. In performing the sensitivity test, three assays were performed for each dilution of the 10-fold diluted standard, and the the relevant content has been added to the manuscript. According to instructions of WOAH-“PRINCIPLES AND METHODS OF VALIDATION OF DIAGNOSTIC ASSAYS FOR INFECTIOUS DISEASES”, a 2-fold dilution experiment below the minimum detection limit was carried out, the results showed that Piadv-1 was detected 6 times and Piadv-2 4 times out of 10 replicates performed, so the detection limits for this test remains 94.8 and 88.4 copies/µL.

10. The authors claim that the method developed for dual detection is satisfactory, but the limit of detection is higher than the LOD of 34.6 copies /µL for PiAdV-2 PCR developed by Chen et al. 2024. This point should be discussed, in addition to the added value of the dual real-time PCR developed by the authors in comparison with the results of Chen et al. 2024.

The author,s answer: The LOD of Chen’s article and the LOD of our experiment both belong to the 101 copies/µL level, and there is no obvious difference. Chen's article focuses on the issue of YPDS and PiAdV-2, which is discussed in the discussion section of the manuscript.

11. Line 194-196. The authors mentioned that the co-infection between PiAdV and other pathogens increase the mortality rate of pigeons. Information about co-infection with PTTV appear later, line 231. This is confusing. Is there a point to discuss about co-infection with other pathogens and PiADV-A and–B infection?

Could the type of PiAdV affect the mortality rate during such co-infection? Is it important to develop a multiplex PCR detecting those pathogens and PiAdV rather than differentiating PiAdV-A and B?

The authors should clarify their point about co-infection with other pathogens and their results.

The author,s answer: Thanks for your comment. The Information about co-infection between PiAdV and other pathogens and co-infection with PTTV has been consolidated in the manuscript. In clinical tests, PiAdV-2 was detected more frequently in dead pigeons than PiAdV-1, but we did not isolate the virus successfully, so we did not conduct animal regression tests to determine their pathogenicity. It is very important to develop a multiplex PCR detecting those pathogens and PiAdV, and we will carry out this work in the next experiment.

12.The authors mentioned a limit of detection of 101 copies/µL for PiAdV-A and B in the abstract, but 94.8 copies for PiAdV-A and 88.4 for PiAdV-B are mentioned in the results and discussion. Indicate the exact LOD for each PiAdV type in the abstract and homogenise in the results. No need for 101, the entire number is more comprehensible.

The author,s answer: Thanks for your advice. The data in the summary and results have been harmonized.

13.Line 55. Can the authors specify if the diagnosis of total PiAdV by PCR is routinely used in breeding farms, in case of pigeon diarrhoea or death?

The author,s answer: In China, pigeon farms usually do not have their own laboratories, and when pigeons suffer from diarrhea or die, the disease material is usually sent to a laboratory that conducts pegion disease research for testing. As for PiAdV-1 or 2 or total PiAdV, it is generally carried out according to the existing detection methods in the laboratory.

14.Line 92. Please specify what type of tissue samples (n=20) from pigeons were tested.

The author,s answer: Thanks for your comment.The type of tissue samples (n=20) from pigeons has been supplemented in the manuscript.

15.Lines 220-221. Is it known if a high morbidity or mortality rate of pigeons in the breeding farm of the regions with the higher PiAdV prevalence was observed?

The author,s answer: The detection rate of PiAdV was high in the dead pigeons from the breeding farm tested by our laboratory. It can be seen that the high mortality or mortality rate of pigeons is correlated with the high prevalence of PiAdV. The relevant description has been added to the manuscript.

16.Line 23. The dual fluorescence PCR method established in this study was used to test 500 pigeon swab samples.

Line 37. The virion has icosahedral symmetry […] the capsid proteins mainly consist of […].

Line 56. “has long detection cycle”, did the authors mean “is time-consuming” ?

Lines 58-59. but these need to the more higher viral load of the sample. This need to be reworded.

Line 138. Delete parenthesis

Line 195. Correct pegions.

The author,s answer: Thanks for your suggestions. The above sentences has been revised in the manuscript

---

## [Decision Letter · Decision Letter 1]

26 May 2025

Dear Dr. Chen,

Thank you for submitting your manuscript to PLOS ONE. After careful consideration, we feel that it has merit but does not fully meet PLOS ONE’s publication criteria as it currently stands. Therefore, we invite you to submit a revised version of the manuscript that addresses the points raised during the review process.

Dear Dr Chen,

Thank you for submitting the revised version of your manuscript, “Establishment and application of a dual real-time PCR assay for differential detection of PiAdV-1 and PiAdV-2 among pigeons in China in 2022-2023”. The paper has now undergone a second round of evaluation.

We invite you to submit a further revision, taking care to address all reviewer comments—particularly those raised by Reviewer 3. Please ensure that the limitations of your study, as well as any additional recommendations, are clearly and thoroughly addressed in the *Conclusion*  section. Specifically, we request that you incorporate a comparative discussion of your findings with recent studies on pigeon aviadenoviruses from China and other countries. Relevant references example:

https://doi.org/10.1002/vms3.662https://doi.org/10.1186/s12917-025-04724-w

Once your revised manuscript is ready, please submit it along with a detailed, point-by-point response to the reviewers’ comments. This will facilitate a prompt and informed final decision.

We look forward to receiving your revised manuscript.

Kind regards,

Ahmed Eisa Elhag

Academic Editor

PLOS ONE

Reviewers' comments:

Reviewer's Responses to Questions

**Comments to the Author**

Reviewer #1: All comments have been addressed

Reviewer #2: All comments have been addressed

Reviewer #3: (No Response)

2. Is the manuscript technically sound, and do the data support the conclusions?

Reviewer #1: Yes

Reviewer #2: Yes

Reviewer #3: No

3. Has the statistical analysis been performed appropriately and rigorously?

Reviewer #1: Yes

Reviewer #2: Yes

Reviewer #3: No

4. Have the authors made all data underlying the findings in their manuscript fully available?

Reviewer #1: Yes

Reviewer #2: Yes

Reviewer #3: No

5. Is the manuscript presented in an intelligible fashion and written in standard English?

Reviewer #1: Yes

Reviewer #2: Yes

Reviewer #3: Yes

**Reviewer #1: ** The authors have adequately addressed my comments raised in a previous round of review and I feel that this manuscript is now acceptable for publication.

**Reviewer #2: ** The manuscript is much clearer now, and the corrections are really helpful. I spotted just two tiny little things that could be improved:

Line 238 : Correct pegions for pigeons

Figure 4 : Correct varient for variant.

Best,

**Reviewer #3: ** The manuscript by Chen et al. concerns the methodology for screening for Pigeon-speciphic Aviadenoviruses in pigeons clinical samples. Adenovirususes are one of the important viral pathogens of domestic pigeons, and for this reason the paper is important. However, despites on corrections made according to the comments of Reviewers 1 and 2, the manuscipt still needs improvements and it can't be published in the present form.

Below is the list of major and minor corrections, that have to been made. After that the paper have to be submitted for another revision stage.

Line 17. Significance of adenoviruses in case of diarrheic diseases of pigeons. Read the latest paper from this topic, provide corrections according to it in the further parts of MS and cite it properly in the manuascript, please: doi.org/10.1016/j.virol.2025.110400.

Introduction section, and pigeon adenoviruses description: This work bases on molecular methods, so provide a brief description of PiAdV genome, please.

Line 53. Spriding of PiAdV in the loft - provide details and reference, please or delete this information.

Line 64. Not true. Virus isolation is not the most common method, but it is the best method in case of proofing the pathogenicity of the virus. Rewrite this section, please.

Line 67. "high viral loads" - What it means "high" - it is too general. Provide some examples, please.

Lines 71-74 - Does it have any practical aspect? Practical means prevention or prophylaxis programms and it's impemmentation in the pigeon flocks?

Line 100. "Pigeon Rotavirus A" (I suppose), because other rotaviruses are also prevalent in pigeons: doi.org/10.1155/tbed/4684235

Provide the strain names used in this study, please.

Line 102. What was the base of this suspicion? Describe it a little bit.

Line 107. How many of each types of the swabs were investigated?

Line 117. cDNA or DNA depending on the virus?

Line 127. Missing space.

Lines 154-158. The section "Sequencing and phylogenetic analysis" .

This section is written very scantly. Provide all sequencing details, please. Provide all details concerning the assembling the sequencing reads into contigs, sequencing errors removing and phylogenetic analysis including method for seletion nucleotide substitution algorithm. Add all information concerning software used for bioinformatic analyzes. Provide the list of accession numbers of generated under this study nucleotide sequences.

Line 195. The description of results of molecular screening for adenoviruses is too general. The statistical analysis of the prevalence of both pigeon aviadenoviruses could improve the results section. I reccomend to add statistical analylis to the methodology.

Lines 208-209. "infected samples" - The samples were positive, not infected.

Lines 2010-2011 - Those results concerns PiAdV-1, PiAdV-2, or both of them?

Table 5. Show results in the table separately for PiAdV-1 and PiAdV-2, please icluding statistical analysis, please.

Line 220. "of was" ??

Fig. 4 and phylogenetic analysis: Add PP999622 and PP999623 sequences to the analysis, please.

Line 243. This is an old data. Currently the RVA is suspected to be a causative agent of YPDS: DOI: 10.1111/tbed.13485.

Lines 250-253. There is one more method previopusly published:

doi.org/10.1016/j.virol.2025.110400 - cite those refereences correctly, please.

The discussion section: Very poor discussion. Too many repetitions of the results, too less real discussion. The results of phylogenetic analysis were not discussed at all.

**Do you want your identity to be public for this peer review?** For information about this choice, including consent withdrawal, please see our Privacy Policy

Reviewer #1: **Yes: ** HARUN ALBAYRAK

Reviewer #2: No

Reviewer #3: No

---

## [Author Response · Author response to Decision Letter 2]

7 Jun 2025

Reviewer #1: The authors have adequately addressed my comments raised in a previous round of review and I feel that this manuscript is now acceptable for publication.

Answer: OK, thanks.

Reviewer #2: The manuscript is much clearer now, and the corrections are really helpful. I spotted just two tiny little things that could be improved:

Line 238 : Correct pegions for pigeons

Figure 4 : Correct varient for variant.

Answer: All corrections have been made

Reviewer #3:

Line 17. Significance of adenoviruses in case of diarrheic diseases of pigeons. Read the latest paper from this topic, provide corrections according to it in the further parts of MS and cite it properly in the manuascript, please: doi.org/10.1016/j.virol.2025.110400.

Introduction section, and pigeon adenoviruses description: This work bases on molecular methods, so provide a brief description of PiAdV genome, please.

Answer: Thanks for your comment. Related contents have been added in the manuscript.

Line 53. Spriding of PiAdV in the loft - provide details and reference, please or delete this information.

Answer: Thanks for your suggestion, and this information has been deleted in the manuscript.

Line 64. Not true. Virus isolation is not the most common method, but it is the best method in case of proofing the pathogenicity of the virus. Rewrite this section, please.

Answer: Thanks for your comment. This section has been rewrited in the manuscript.

Line 67. "high viral loads" - What it means "high" - it is too general. Provide some examples, please.

Answer: Thanks for your advice. This section has been redescribed in the manuscript.

Lines 71-74 - Does it have any practical aspect? Practical means prevention or prophylaxis programms and it's impemmentation in the pigeon flocks?

Answer: Thanks for your comment. Through rapid clinical diagnosis, early detection, early isolation and early treatment can be achieved, reducing the loss of pigeon flocks. The relevant content has been added in the manuscript.

Line 100. "Pigeon Rotavirus A" (I suppose), because other rotaviruses are also prevalent in pigeons: doi.org/10.1155/tbed/4684235. Provide the strain names used in this study, please.

Answer: Thanks for your comment. I am sorry for the mistake, and the RVA strain was used in this study. Our laboratory only has RVA and no other RVs. We have made revisions in the manuscript.

Line 102. What was the base of this suspicion? Describe it a little bit.

Answer: Thanks for your comment. Liver tissue treatment was performed on 20 pigeons with clinical signs such as vomiting and diarrhea.

Line 107. How many of each types of the swabs were investigated?

Answer: Oropharyngeal swab and cloacal swab of one pigeon mixed in one tube

Line 117. cDNA or DNA depending on the virus?

Answer: The mistake has been corrected in the manuscript.

Line 127. Missing space.

Answer: All corrections have been made.

Lines 154-158. The section "Sequencing and phylogenetic analysis" .

This section is written very scantly. Provide all sequencing details, please. Provide all details concerning the assembling the sequencing reads into contigs, sequencing errors removing and phylogenetic analysis including method for seletion nucleotide substitution algorithm. Add all information concerning software used for bioinformatic analyzes. Provide the list of accession numbers of generated under this study nucleotide sequences.

Answer: Thanks for your comment. This content was supplemented with details in the manuscript.

Line 195. The description of results of molecular screening for adenoviruses is too general. The statistical analysis of the prevalence of both pigeon aviadenoviruses could improve the results section. I reccomend to add statistical analylis to the methodology.

Answer: Thanks for your comment. Appropriate additions has been added n the manuscript.

Lines 208-209. "infected samples" - The samples were positive, not infected.

Answer: Thanks for your comment. The mistake has been corrected in the manuscript.

Lines 2010-2011 - Those results concerns PiAdV-1, PiAdV-2, or both of them?

Answer: Thanks for your comment. The content in doubt has been revised in the manuscript.

Table 5. Show results in the table separately for PiAdV-1 and PiAdV-2, please icluding statistical analysis, please.

Answer: Thanks for your advice. The content has been revised in the manuscript.

Line 220. "of was" ??

Answer: The mistake has been revised in the manuscript.

Fig. 4 and phylogenetic analysis: Add PP999622 and PP999623 sequences to the analysis, please.

Answer: The content has been added in the manuscript.

Line 243. This is an old data. Currently the RVA is suspected to be a causative agent of YPDS: DOI: 10.1111/tbed.13485.

Answer: The content has been revised in the manuscript.

Lines 250-253. There is one more method previopusly published:

doi.org/10.1016/j.virol.2025.110400 - cite those refereences correctly, please.

Answer: Thanks for your comment. The content has been added in the manuscript.

The discussion section: Very poor discussion. Too many repetitions of the results, too less real discussion. The results of phylogenetic analysis were not discussed at all.

Answer: Thanks for your comment. We have made modifications to the discussion section , and added the phylogenetic analysis in the manuscript

---

## [Editor Report · Decision Letter 2]

1 Aug 2025

Establishment and application of a dual real-time PCR assay for differential detection of PiAdV-1 and PiAdV-2 among pigeons in China in 2022-2023

PONE-D-24-53219R2

Dear Dr. Chen,

We’re pleased to inform you that your manuscript has been judged scientifically suitable for publication and will be formally accepted for publication once it meets all outstanding technical requirements.

Kind regards,

Ahmed Eisa Elhag

Academic Editor

PLOS ONE

Additional Editor Comments (optional):

Dear Dr. Chen,

Thank you for submitting the revised version of your manuscript titled “Establishment and application of a dual real-time PCR assay for differential detection of PiAdV-1 and PiAdV-2 among pigeons in China in 2022–2023.”

We appreciate your patience and the efforts you have made to enhance the manuscript and address the reviewers’ comments after the second round of evaluation.

Before we proceed with the final acceptance, we kindly ask that the manuscript be thoroughly proofread by a native English speaker or a professional with expertise in scientific writing. There are still some language issues and typographical errors that need attention.
---

## [Editor Report · Acceptance letter]

PONE-D-24-53219R2

PLOS ONE

Dear Dr. Chen,

I'm pleased to inform you that your manuscript has been deemed suitable for publication in PLOS ONE. Congratulations! Your manuscript is now being handed over to our production team.

Kind regards,

on behalf of

Dr. Ahmed Eisa Elhag

Academic Editor

PLOS ONE